# Biological Activity of Pseudovitamin B_12_ on Cobalamin-Dependent Methylmalonyl-CoA Mutase and Methionine Synthase in Mammalian Cultured COS-7 Cells

**DOI:** 10.3390/molecules25143268

**Published:** 2020-07-17

**Authors:** Tomohiro Bito, Mariko Bito, Tomomi Hirooka, Naho Okamoto, Naoki Harada, Ryoichi Yamaji, Yoshihisa Nakano, Hiroshi Inui, Fumio Watanabe

**Affiliations:** 1Department of Agricultural, Life and Environmental Sciences, Faculty of Agriculture, Tottori University, Tottori 680-8553, Japan; watanabe@tottori-u.ac.jp; 2Division of Applied Life Sciences, Graduate School of Life and Environmental Sciences, Osaka Prefecture University, Sakai, Osaka 599-8531, Japan; cobalamin1985@yahoo.co.jp (M.B.); hiro.207.tomo@gmail.com (T.H.); harada@biochem.osakafu-u.ac.jp (N.H.); yamaji@biochem.osakafu-u.ac.jp (R.Y.); nakano@biochem.osakafu-u.ac.jp (Y.N.); 3The United Graduate School of Agricultural Sciences, Tottori University, Tottori 680-8553, Japan; d18a2005y@edu.tottori-u.ac.jp; 4Department of Nutrition, College of Health and Human Sciences, Osaka Prefecture University, Habikino, Osaka 583-8555, Japan; inui@biochem.osakafu-u.ac.jp

**Keywords:** adenyl cobamide, mammalian cell, methionine synthase, methylmalonyl-CoA mutase, pseudovitamin B_12_, transcobalamin II, vitamin B_12_

## Abstract

Adenyl cobamide (commonly known as pseudovitamin B_12_) is synthesized by intestinal bacteria or ingested from edible cyanobacteria. The effect of pseudovitamin B_12_ on the activities of cobalamin-dependent enzymes in mammalian cells has not been studied well. This study was conducted to investigate the effects of pseudovitamin B_12_ on the activities of the mammalian vitamin B_12_-dependent enzymes methionine synthase and methylmalonyl-CoA mutase in cultured mammalian COS-7 cells to determine whether pseudovitamin B_12_ functions as an inhibitor or a cofactor of these enzymes. Although the hydoroxo form of pseudovitamin B_12_ functions as a coenzyme for methionine synthase in cultured cells, pseudovitamin B_12_ does not activate the translation of methionine synthase, unlike the hydroxo form of vitamin B_12_ does. In the second enzymatic reaction, the adenosyl form of pseudovitamin B_12_ did not function as a coenzyme or an inhibitor of methylmalonyl-CoA mutase. Experiments on the cellular uptake were conducted with human transcobalamin II and suggested that treatment with a substantial amount of pseudovitamin B_12_ might inhibit transcobalamin II-mediated absorption of a physiological trace concentration of vitamin B_12_ present in the medium.

## 1. Introduction

Vitamin B_12_ or cobalamin (Cbl) is commonly known as the red-colored vitamin [1]. Cbl has a cobalt-coordinated nucleotide, which provides its base (5,6-dimethylbenzimidazole) as the lower axial ligand. The inactive vitamin, usually CN-Cbl or OH-Cbl, is readily converted into the coenzyme forms methylcobalamin (MeCbl) (see Figure 1), a coenzyme of methionine synthase (MS) (EC 2.1.1.13) involved in methionine biosynthesis [1], and 5′-deoxyadenosylcobalamin (AdoCbl), a coenzyme of methylmalonyl-CoA mutase (MCM) (EC 5.4.99.2) involved in amino acid and odd-chain fatty acid metabolism in mammalian cells [2,3]. During Cbl deficiency, holo-MCM is significantly reduced and the stable apoenzyme is accumulated [4]. In contrast, nearly no apo-MS is observed at any Cbl-status, because the apoenzyme is extremely labile [5]. Supplementation with Cbl induces the translational up-regulation of MS [5], immediately followed by saturation and stabilization of the freshly produced MS-molecules.

Human beings have a complex system for gastrointestinal absorption of dietary Cbl [6], which is released from food proteins and then bound to a salivary Cbl-binding protein, haptocorrin, in the stomach. The protein moiety of the haptocorrin–Cbl complex is digested by pancreatic proteases in the duodenum, and then, free Cbl is formed. The released Cbl binds to a gastric Cbl-binding protein, the intrinsic factor (IF), in the proximal ileum. The resulting IF-Cbl complex is internalized into mucosal cells of the distal ileum via receptor-mediated endocytosis. After the digestion of the protein moiety of the IF-Cbl complex in the mucosal cells, the released free Cbl moves into the blood circulation and binds to transcobalamin II (TCII).

Cbl is synthesized by certain bacteria and archaea but not by plants. It accumulates in animal tissues through microbial interactions in the food chain [7]. For example, ruminants, such as cattle and sheep, acquire Cbl through a symbiotic relationship with the bacteria in their stomachs [7]. Foods obtained from animals (meat, milk, and fish) are major dietary sources of Cbl for humans [8]. Beside Cbl, various bacteria produce a substantial amount of adenyl cobamide [(Ade)Cba] known as pseudovitamin B_12_ [9,10,11]. Studies have reported that human feces [12], sheep duodenal digesta [13], and dietary supplements from cyanobacteria [7] contain considerable quantities of (Ade)Cba. It has been demonstrated that ingestion of (Ade)Cba-containing food does not disturb the gastrointestinal absorption of Cbl, because the apparent affinity constant for the binding of (Ade)Cba to human IF is very low [14]. Moreover, infusion of large quantities of (Ade)Cba into Cbl-sufficient rats has no consequence on Cbl-related metabolism [15].

The structural specificity of the base moieties of Cba on the activities of Cbl-dependent enzymes is not well studied. The in vitro binding affinity of bacterial and mammalian recombinant MCM for various Cba species varies significantly depending on the bases in the lower ligand and, for instance, mammalian MCM does not complex with (Ade)Cba [16,17]. In contrast to MCM, mammalian MS has low specificity for alterations in the base moiety and (Ade)Cba is fully activated and functions as a cofactor for MS in vitro [18]. However, the effect of (Ade)Cba on the activities of Cbl-dependent enzymes has not been studied using cultured mammalian cells.

In the present study, we investigated the effect of (Ade)Cba addition on the activities of MCM and MS in cultured mammalian cells in the presence or absence of the specific transporter TCII and also determined, whether (Ade)Cba functions as a coenzyme or an inhibitor of mammalian MCM and MS.

## 2. Results and Discussion

### 2.1. Effects of (Ade)OH-Cba on MS Activity in a Homogenate of Mammalian Cells Grown with or without OH-Cbl

After culturing COS-7 cells in media with or without the addition of OH-Cbl, the activities of holo- and apo-MS were assayed in the cell homogenate using the reaction mixture with and without OH-Cbl, respectively. Low holo-MS activity was observed in the homogenate of COS-7 cells grown under control conditions (without OH-Cbl) (Table 1). This was because only a trace amount of Cbl (approximately 10 pmol/L), necessary for cell survival was present in the fetal bovine serum (FBS) used. However, in the homogenates of cells grown with 0.1 and 1 µM OH-Cbl, the holo-enzyme activity was increased considerably. The total MS activity was measured in all cases after adding excess (50 µM) OH-Cbl to the reaction mixture. All recorded rates matched the original holo-enzyme activity (without the added OH-Cbl) indicating that the majority of MS exists in cells as a holo-enzyme, whereas the apoenzyme is generally absent (making impossible further activation of MS by external Cbl). These observations were consistent with previously reported results [19,20]. Because the maximal enzyme activity was found at the addition of 1 µM OH-Cbl, (Ade)OH-Cba was also added at the concentration of 1 µM to the growth medium. The addition of (Ade)OH-Cba had no effect on the activity of holo-MS enzyme in the homogenates of cells grown at various concentrations of the external OH-Cbl (0, 0.1, and 1 µM), besides the endogenous Cbl (10 pM) in the medium.

OH-Cbl functions an internal ribosome entry-site transactivating factor and activates the 5′-upstream region of MS mRNA to induce MS mRNA translation [21]. In contrast, (Ade)OH-Cba could not induce the expression of MS (Figure 2). Presumably, (Ade)OH-Cba does not have the ability to associate with the internal ribosome entry site.

### 2.2. Effects of OH-Cbl and (Ade)OH-Cba, Added to the Growth Medium, on MS Activity in a Homogenate of MS cDNA-Transfected Mammalian Cells

MS cDNA- or mock-transfected cells were cultured with or without 1 µM OH-Cbl or (Ade)OH-Cba for 2 days. Very low native holo-MS activity was detected in the homogenates of mock- and MS cDNA-transfected cells grown without OH-Cbl (Table 2). The addition of OH-Cbl to the MS-reaction mixture significantly increased MS activity in the homogenate of MS cDNA-transfected cells, but not in the mock-cell homogenate (Table 2). These results indicate that the synthesis of MS apoenzyme was noticeably induced in MS cDNA-transfected cells. The cultivation of cells at 1 µM OH-Cbl significantly increased the holo-enzyme activity, which was identical to the total enzyme activity, indicating that all MS enzymes existed as a holo-enzyme in the cells. Moreover, the addition of 1 µM OH-Cbl to cell medium significantly increased the total MS activity in MS cDNA-transfected cells, suggesting that OH-Cbl might induce synthesis of MS protein even in the transfected cells. Contrary to the above findings, (Ade)OH-Cba did not affect the total activity in MS of cDNA-transfected cells, suggesting that (Ade)OH-Cba could not induce synthesis of MS protein. Figure 3 shows the effects of cDNA transfection and addition of OH-Cbl or (Ade)OH-Cba on MS expression in the cells. Although MS protein was not detected in mock-transfected cells because of its trace amount in the sample preparation, an immunoreactive band associated with the exogenous MS protein was detected in MS cDNA-transfected cells. In fact, OH-Cbl-treated MS cDNA-transfected cells showed two immunoreactive bands whose molecular masses were identical to exogenous (from transfected cDNA) and endogenous MS proteins upper and lower bands, respectively. Higher molecular mass of exogenous MS protein was due to the FLAG-tag in the fusion construct. The addition of 1 µM (Ade)OH-Cba did not induce endogenous MS expression and did not affect the production of exogenous MS. Moreover, OH-Cbl did not affect exogenous MS either.

The addition of 1 µM (Ade)OH-Cba to the growth medium increased the holo-enzyme activity in homogenates of transfected cells by approximately 42% of that obtained with the addition of 1 µM OH-Cbl. The total enzyme activity was approximately 1.6 times higher than the holo-enzyme activity, suggesting that approximately 60% of total enzyme activity was derived from the holo-enzyme. These results suggest that (Ade)OH-Cba can function as a coenzyme of MS in cultured cell systems, but this Cbl-analogue has lower activity compared with OH-Cbl.

### 2.3. Effects of Various Concentrations of OH-Cba or (Ade)OH-Cba on the Enzyme Activity of apo-MS

As described above, apo-MS was significantly induced in MS cDNA-transfected cells, grown in the absence of OH-Cbl. Therefore, we used this cell homogenate as a source of MS apoenzyme for subsequent experiments. To evaluate whether MS can efficiently use (Ade)OH-Cba as the coenzyme, apparent *K*_d_ values of the apoenzyme were determined in a cell-free system. The apparent *K*_d_ values for OH-Cbl and (Ade)OH-Cba were estimated to be 0.4 and 26.3 µM, respectively (Figure 4), indicating that (Ade)OH-Cba has a significantly lower affinity for MS (approximately 1/65) than OH-Cbl.

### 2.4. Effects of (Ade)OH-Cba on MCM Activity in a Homogenate of Mammalian Cells Grown in the Presence or Absence of OH-Cbl

A similar set of experiments was performed to analyze the activity of another Cbl-dependent enzyme MCM. After growing COS-7 cells in a medium with or without the addition of OH-Cbl, the activities of holo- and apo-MCM were assayed in the cell homogenate using the reaction mixture with and without AdoCbl, respectively (Table 3). The holo-enzyme activity was considerably increased in the homogenates of cells grown with 0.1 and 1.0 µM OH-Cbl. However, a higher total activity of MCM (apo + holo) was detected in the homogenates of control cells (no addition of OH-Cbl) and then the activity gradually decreased by approximately 30% of the total MCM activity found in cells supplemented with 1 µM OH-Cbl. These results suggest that the cells grown under control conditions apparently develop severe Cbl deficiency, characterized by expression of high levels of apo-MCM. Such phenomena coincided with previously described results [4].

As shown in Table 3, the addition of 1 µM (Ade)OH-Cba had no effect on the total MCM activity in the homogenates of cells grown with various concentrations (0, 0.1, and 1 µM) of OH-Cbl, beside the endogenous 10 pM Cbl in the medium. In addition, (Ade)OH-Cba had little effect on holo-MCM activity, although a small (7%) decrease was observed in cells supplemented with 0.1 µM OH-Cbl.

### 2.5. Effects of (Ade)Ado-Cba on the Activity of apo-MCM in Mammalian Cells

As described above, the majority of MCM was derived from the apoenzyme in the homogenate of cells grown under control conditions (no addition of OH-Cbl). This cell homogenate was used as a source of MCM apoenzyme for subsequent experiments. To determine whether (Ade)Ado-Cba can function as a coenzyme of MCM, we investigated the effects of (Ade)Ado-Cba on the activity of apo-MCM (Table 4). The enzyme activity significantly increased with the increasing amount of AdoCbl. The addition of 33 µM (Ade)Ado-Cba had no effect on the enzyme activity in the presence or absence of 3.3 µM AdoCbl. These results suggest that (Ade)Ado-Cba does not function as a coenzyme or inhibitor of MCM.

### 2.6. Effects of TCII on holo-MCM Activity in a Homogenate of Mammalian Cells Grown in the Presence of OH-Cbl

After the addition of 1 nM TCII to the medium, the cells were grown for 48 h, collected, and homogenized, and then the holo-MCM activity was assayed (Table 5). In the absence of TCII, no enzyme activity was detected with the addition of 1 nM OH-Cbl to the growth medium. However, the addition of a higher concentration (1 µM OH-Cbl) to the growth medium significantly stimulated the enzyme activity (2.1 ± 0.10 nmol/min/mg protein), irrespective of presence or absence of TCII. Approximately 67% of the aforementioned “maximal” enzyme activity was detected in the homogenate of cells grown in the presence of 1 nM TCII and 1 nM OH-Cbl. Using TCII and OH-Cbl, each at 1 nM, the effects of various concentrations of (Ade)OH-Cba on the activity of holo-MCM were investigated. It was observed that the enzyme activity gradually decreased with increasing concentration of (Ade)OH-Cba and reduced by approximately 43% with the addition of 10 nM (Ade)OH-Cba. Considering that the addition of (Ade)Cba had no effect on MCM activity in the absence of TCII, these results suggest that (Ade)Cba might competitively inhibit the TCII-mediated Cbl uptake under the experimental conditions.

### 2.7. Biological Properties of (Ade)Cba in Mammalian Cells

We characterized the biological activity of (Ade)Cba as a cofactor of MS and MCM in mammalian cells. (Ade)OH-Cba did not act as either a coenzyme or an inhibitor for MCM in COS-7 cells (Table 3 and Table 4). These results are supported by previous studies, which report that small changes in the base moiety of the lower ligand of Cba noticeably affect the binding of Cba to bacterial and mammalian MCM, which shows very low affinity for (Ade)Ado-Cba [16,17]. Yet, the bound (Ade)Cba can act as a cofactor of MS and MCM in vitro, and it can activate and function as a coenzyme of MS in the mammalian cell system. In our study, (Ade)Cba had a significantly lower affinity for the recombinant human MS than OH-Cbl. However, using an MS apoenzyme purified from the human placenta, Kolhouse et al. [18] reported that the apoenzyme has a higher affinity for (Ade)OH-Cba than OH-Cbl and that (Ade)OH-Cba is fully activated and functional as a cofactor for MS. As shown in Table 2, approximately 40% apoenzyme was formed by treating MS cDNA-transfected cells with 1 µM (Ade)OH-Cba but not with 1 µM OH-Cbl, suggesting that (Ade)OH-Cba cannot fully function as a coenzyme for MS in this cell culture system. The rationale for lacking activation comes from the saturation curves, which indicate that (Ade)OH-Cba has a significantly lower affinity for MS than OH-Cbl (Figure 4). Currently, it is unclear why contradictory results were obtained in the current study and ref. [18]. The exogenously expressed recombinant FLAG-tagged MS possibly behaves differently than native MS. During enzymatic reactions, we reduced Cbl/Cba by titanium citrate, which usually gives [Co^1+^] as the predominant cofactor form in the reaction medium. In contrast, Kolhouse at al employed reduction by mercaptoethanol, where the major steady-state form under reduction-oxidation is [Co^2+^] [18]. The oxidative state of Co-ion inside the corrin ring affects coordination patterns of all corrinoids in respect to the surrounding ligands, which might account for the reported differences. Further studies are needed to clarify this issue.

On the other hand, it has been reported that small changes in the base moiety of the lower ligand of Cba affected the binding affinity of Cba to bacterial and mammalian MCM [16,17]. However, (Ade)Ado-Cba could not function as a coenzyme or an inhibitor of MCM in this mammalian system. In this regard, our study fully agrees with the literature.

During Cbl deficiency, a significant quantity of a stable MCM apoenzyme is accumulated in mammalian cells. In this case, (Ade)Cba would neither function as a coenzyme of MCM nor induce the expression of MS mRNA in the cells. Yet, the activity of MS in vivo might decrease significantly, even though (Ade)Cba can fully function as a coenzyme. (Ade)Cba can bind TCII at a similar [22] or lower affinity than Cbl [14]. Our results (Table 5) suggest that a substantially higher concentration of 10 nM (Ade)OH-Cba inhibits the TCII-mediated uptake of 1 nM Cbl in mammalian cells. These results suggest that if a substantial amount of (Ade)Cba is absorbed in the intestine, it may accelerate the development of Cbl deficiency during Cbl-deprived conditions. Bacteria or their products cross the intestinal barrier via bacterial translocation [23,24]. Moreover, alcohol and stress increase the membrane permeability of the intestinal tract [25,26]. Thus, (Ade)Cba synthesized by intestinal bacteria and/or ingested from edible cyanobacteria can penetrate the intestinal wall and enter the bloodstream.

The consequences of such penetration for Cbl-status are, however, questionable. Thus, the administration of large quantities of (Ade)OH-Cba by subcutaneous infusion into rats on a Cbl-sufficient diet did not affect Cbl-related metabolism [15], suggesting that even substantial amounts of (Ade)Cba do not inhibit mammalian TCII-mediated Cbl uptake under Cbl-sufficient conditions. Consequently, small amounts of (Ade)Cba, which enters the bloodstream via the intestinal wall may not be harmful for humans who ingest sufficient amounts of Cbl.

## 3. Materials and Methods

### 3.1. Materials

The following reagents were used in this study: 5′-deoxyadenosylcobalamin, hydoroxocobalamin, methylmalonyl-CoA tetralithium salt hydrate, 5-methyltetrahydrofolic acid disodium salt, 5′-iodo-5′-deoxyadenosine, and Dulbecco’s modified Eagle’s medium (DMEM) (Sigma-Aldrich, St. Louis, MO, USA); cyanocobalamin (FUJIFILIM Wako Pure Chemical Corp., Osaka, Japan); Cosmosil 5C18-AR-II HPLC column (Nacalai Tesque Inc., Kyoto, Japan); Ni-sepharose (GE Healthcare UK Ltd., Buckinghamshire, England); Gibco fetal bovine serum (FBS; Thermo Fisher Scientific, Waltham, MA, USA); PrimeStar HS DNA polymerase (TAKARA Bio Inc., Shiga, Japan); and polymerase chain reaction (PCR) purification kit (Qiagen, Hilden, Germany). Other chemicals were used as highly purified reagents.

### 3.2. Preparation of (Ade)CN-Cba, (Ade)Ado-Cba, and (Ade)OH-Cba

(Ade)CN-Cba was extracted from the cultured cells of *Propionibacterium acidipropionici* JCM6427 (RIKEN BioResource Research Center, Tsukuba, Ibaraki, Japan), purified to homogeneity, and identified according to the method described by Tanioka et al. [27]. (Ade)CN-Cba solution was bubbled with N_2_ gas for 20 min, reduced with NaBH_4_, and neutralized with 1 M HCl. Most of the (Ade)CN-Cba was converted into (Ade)OH-Cba. After desalting the treated solution with a Sep-pak Vac 20 cc (5 g) C18 cartridge (Waters Corp, Milford, MA, USA), the resulting (Ade)OH-Cba was separated from trace unreacted (Ade)CN-Cba with a silica gel 60 thin-layer chromatography (TLC) under the same conditions as described previously [28]. For the preparation of (Ade)Ado-Cba, a small amount of 5′-iodo-5′-deoxyadenosine was added to (Ade)CN-Cba that had been treated with N_2_ gas and NaBH_4_. The formed (Ade)Ado-Cba was isolated with a silica gel 60 TLC under the same conditions.

### 3.3. Enzyme Assays

For enzyme assays, the cells were suspended in 10 mmol/L potassium phosphate buffer (pH 7.0) containing 10% (*w/v*) sucrose, disrupted using an ultrasound disintegrator, and then centrifuged at 12,000× *g* for 20 min at 4 °C. The supernatant was used as the crude enzyme. The activities of MS and MCM as Cbl-dependent enzymes were assayed using the HPLC method as described previously [29,30]. In brief, total enzyme (holo-enzyme and apo-enzyme) and holo-enzyme activities were determined with or without Cbl as coenzymes (AdoCbl for MCM and MeCbl for MS). In the assay of MCM activity, succinyl CoA produced by the enzymatic reaction of crude enzyme and methylmalonyl CoA as a substrate was assayed by measuring absorbance at 254 nm. In the assay of MS activity, tetrahydrofolic acid formed by the enzymatic reaction of crude enzyme and 5′-methyl tetrahydrofolic acid as a substrate was assayed by measuring the fluorescence intensity at an excitation wavelength of 290 nm and emission wavelength of 356 nm. The apparent *K*_d_ values of MS apoenzyme for OH-Cbl and (Ade)OH-Cba were calculated using a curve fitting software (Microsoft Excel Solver [31]). Data were plotted using the following equation: B/Bmax = [Cbl]/(*K*_d_ + [Cbl]), B = specific activity of MS at various concentrations of OH-Cbl or (Ade)OH-Cba, Bmax = maximum specific activity of MS upon addition of OH-Cbl or (Ade)OH-Cba, and [Cbl] = concentration (µM) of OH-Cbl or (Ade)OH-Cba.

### 3.4. Construction of Vector for Intracellular Expression System of Human MS and Human TCII

First, the cDNA fragments encoding human MS (hMS) and human TCII (hTCII) containing UTR were amplified from Human Marathon-Ready cDNA (brain, cerebral cortex; Clontech, CA, USA) by PCR using the following primer sets: hMS-F1 (5′-AGCCAACGGGAGGCGTCAAAAGACC-3′) and hMS-R1 (5′-CAGGAAGACCCTGCTCCTCTACAAGG-3′), and hTCII-F1 (5′-GGAGTCTTTCCCGATTCTTGCT-3′) and hTCII-R1 (5′-ACCACAGAACGAGTGGTCTTCA-3′). To obtain the open reading frame of hMS and hTCII, each first PCR product as a template was amplified using the following primer sets: hMS-F2 (5′-GGGATCCTCACCCGCGCTCCAAGACCTG-3′) and hMS-R2 (5′-GCACCCTAGGTTGTATTTCCTTGAGG-3′), and hTCII-F2 (5′-GGGATCCGCCATGAGGCACCTTGGGGCCTTCC-3′) and hTCII-R2 (5′-GGATCCCCAGCTAACCAGCCTCAGCTC-3′). The DNA fragments generated by PCR were purified by gel electrophoresis and ligated into pCR2.1-TOPO-TA. The resulting constructions were designated as pCR/hMS and pCR/hTCII, respectively. The sequences of the inserted region in pCR2.1-TOPO-TA were verified by the dideoxy chain primer method.

The pCR/hMS vectors that were digested with *Bgl* II/*Eco* RI DNA fragments were purified by gel electrophoresis and then ligated into p3xFLAG-Myc-CMV that was digested with the same restriction enzymes and then transformed into *Escherichia coli* strain DH5α. The resulting constructions were designated as p3xFLAG-hMS_1892_. The pCR/hMS vectors that were digested with *Spe* I/*Xba* I DNA fragments were purified by gel electrophoresis and then ligated into p3xFLAG-hMS_1892_ that was digested with *Xba* I restriction enzyme and then transformed into *E. coli* strain DH5α. The resulting constructions were designated as p3xFLAG-hMS. The pCR/hTCII vectors that were digested with *BamH* I DNA fragments were purified by gel electrophoresis and then ligated into pTRE2hyg-hGAPDH-Myc-His that was digested with the same restriction enzymes. The resulting constructions were designated as pTRE2hyg-hTCII (KozBam)-Myc-His.

### 3.5. Cell Culture

COS-7 (African green monkey kidney) cells (Clontech, Mountain View, CA, USA) were cultured in 10 mL DMEM containing 10% (*v/v*) FBS, penicillin (100 U/mL), and streptomycin (100 μg/mL) on a 100-mm dish at 37 °C in a humidified 5% CO_2_ and 95% air (*v/v*) atmosphere. Authentic OH-Cbl was added into the medium at various concentrations (0, 0.1, and 1.0 μmol/L) in the presence or absence of 1.0 μmol/L (Ade)OH-Cba. After reaching confluence, the cells were harvested and homogenized in 100 mmol/L potassium phosphate buffer (pH 7.0) containing leupeptin (1 mg/mL), aprotinin (10 mg/mL), and 0.1 mmol/L 4-(2-aminoethyl)-benzenesulfonyl fluoride, in a Teflon homogenizer at 4 °C. The homogenate was centrifuged at 10,000× *g* for 10 min at 4 °C, and the supernatant fraction was used as a crude enzyme solution for the enzyme assay.

HEK 293 (tet-on) (human embryonic kidney, tetracycline-induced expression system) cells were cultured in 10 mL DMEM containing FBS, penicillin, streptomycin, G418 (100 μg/mL), and NaHCO_3_ (1 g/L) on a collagen-coated dish (100 mm) at 37 °C in a humidified 5% CO_2_ and 95% air (*v/v*) atmosphere.

### 3.6. Transfection of hMS Plasmid to COS-7 Cells and hTCII Plasmid to HEK293 Cells

COS-7 cells (6.0 × 10^6^ cells) were incubated in a total volume of 500 µL of phosphate-buffered saline containing 16 µg of hMS plasmid for 10 min on ice before electroporation. The cells were electroporated using GenePulser Xcell electroporator (220 V, 950 µF, Bio-Rad) and then incubated on ice for 10 min. After the addition of 500 µL DMEM (serum- and antibiotics-free), the cells were incubated at room temperature for 10 min. The cell solution was diluted to 9 mL in DMEM and then cultivated for 48 h.

The hTCII plasmids were transfected into HEK293 (tet-on) cells (Clontech, Mountain View, CA, USA) using Lipofectamine 2000 (Invitrogen; Thermo Fisher Scientific, Waltham, MA, USA), following the manufacturer’s instructions. Briefly, DNA (2 µg) was mixed with 4 µL Lipofectamine 2000 in 200 μL DMEM (not containing FBS and antibiotics) and incubated under room temperature for 15 min. After incubation, the DNA mixture was added into a 35-mm culture dish at a cell density of around 60% confluency and incubated at 37 °C for 5 h. After 24 h, the cells were replaced in DMEM containing FBS, penicillin, streptomycin, G418, and hygromycin (200 µg/mL) and grown until colony formation for approximately 2 weeks. The medium was exchanged every 3 days.

### 3.7. Purification of Recombinant Human TCII from HEK293 Cells

The HEK293 cells transfected hTCII were grown until confluence, replaced with DMEM containing 10 µg/mL doxycycline (not containing FBS, penicillin, streptomycin, and G418), and then incubated at for 24 h at 37 °C. The culture supernatant was collected, to which 20 mmol/L sodium phosphate buffer (pH 8.0) containing 20 mmol/L imidazole was added and incubated at 4 °C for 1 h with Ni-sepharose resin that equilibrated with 20 mmol/L sodium phosphate buffer (pH 8.0) containing 20 mmol/L imidazole and 150 mmol/L NaCl as a binding buffer. After incubation, the resin was washed with the binding buffer and then eluted with 20 mmol/L sodium phosphate buffer (pH 8.0), which contained 300 mmol/L imidazole and 150 mmol/L NaCl as an elution buffer. The elution was concentrated with Amicon Ultra-4 unit (Merck) as human TCII.

### 3.8. Sodium Dodecyl Sulfate Polyacrylamide Gel Electrophoresis (SDS-PAGE) and Western Blot Analysis

Harvested cells were disrupted with a sonicator in 100 µL of 50 mM Tris-HCl buffer (pH 7.5) containing 150 mM NaCl, 0.5% (*v/v*) Nonidet P-40, 10 mM sodium pyrophosphate, 2 mM EDTA, 10 mg/mL leupeptin, 1.0 mg/mL aprotinin, and 1 mM phenylmethylsulfonyl fluoride. The cell homogenate was centrifuged at 20,000× *g* for 20 min at 4 °C, and the supernatant was used as a crude enzyme for Western blot analysis. The crude enzyme was treated with SDS-PAGE according to the Laemmli method [32]. After electrophoresis, the proteins were transferred onto a polyvinylidene difluoride (PVDF) membrane (Immuno-Blot PVDF; Bio-Rad) in a Trans-Blot SD semidry electrophoretic transfer cell (Bio-Rad). Nonspecific binding was blocked using 6% (*w/v*) nonfat dry milk in phosphate-buffered saline. The PVDF membranes were probed with goat polyclonal anti-MTR antibody (ab9209, Abcam) or rabbit polyclonal anti-c-Myc (A-14) antibody (Santa Cruz Biotechnology, Dallas, TX, USA). Immunoreactive components were visualized with an anti-rabbit IgG (H + L) secondary antibody (Bio-Rad) coupled to horseradish peroxidase, tetramethylbenzidine, Super Signal West Pico (PIEROE) or Immobilon Western Chemiluminescent HRP Substrate (Merck), according to the manufacturer’s instructions. Chemiluminescence was detected by LAS 4000 (Fuji Film, Tokyo, Japan). A protein ladder (Broad) was used to determine the molecular mass.

### 3.9. Protein Quantitation

Protein assay was conducted according to the method of Bradford [33] using bovine serum albumin as a standard.

### 3.10. Statistical Analysis

The result shown in Exp. 2 of Table 5 was analyzed by one-way ANOVA with Bonferroni’s post-hoc test using GraphPad Prism 4 (GraphPad Software, La Jolla, CA, USA). Other data were analyzed using Student’s t-test for pairwise comparison.

## Figures and Tables

**Figure 1 molecules-25-03268-f001:**
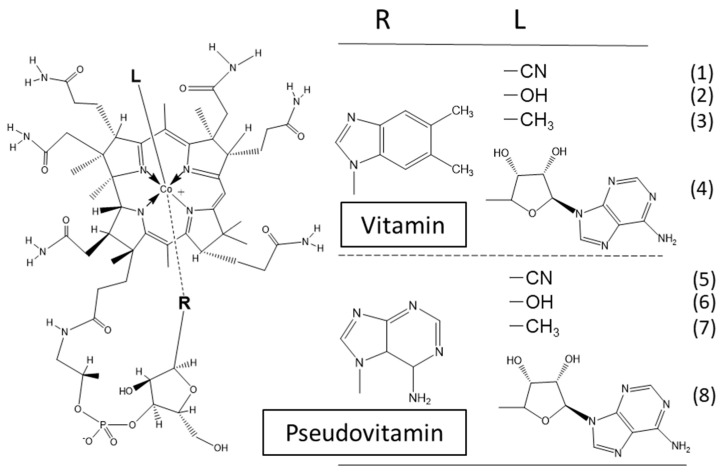
Structural formula of vitamin B_12_ and partial structures of vitamin B_12_-related compounds. (**1**) Cyanocobalamin (vitamin B_12_, Cbl, or CN-Cbl), (**2**) hydroxocobalamin (OH-Cbl), (**3**) methylcobalamin (MeCbl), (**4**) adenosylcobalamin (AdoCbl), (**5**) pseudovitamin B_12_ [(Ade)Cba], (**6**) hydroxo-form of pseudovitamin B_12_ [(Ade)OH-Cba], (**7**) methyl-form of pseudovitamin B_12_, [(Ade)Me-Cba], and (**8**) adenosyl form of pseudovitamin B_12_ [(Ade)Ado-Cba].

**Figure 2 molecules-25-03268-f002:**
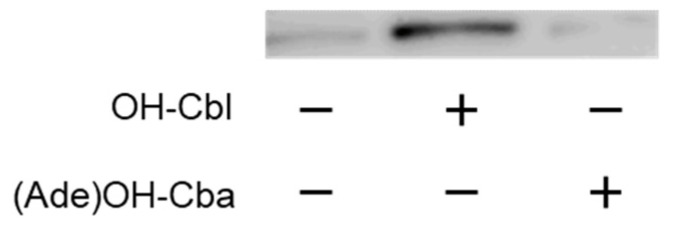
Western blotting of MS protein in the homogenate of COS-7 cells treated with or without (Ade)OH-Cba in the presence or absence of OH-Cbl. Cells were cultured in the medium containing OH-Cbl at 0 or 1 µM with or without 1 µM (Ade)OH-Cba for 2 days. The cell homogenate was centrifuged at 20,000× *g* for 20 min at 4 °C, and the supernatant was used as a source of the crude enzyme for Western blotting. MS protein on polyvinylidene difluoride (PVDF) membranes was visualized with goat polyclonal anti-MTR antibody. Data show typical immunoreactive patterns of the MS protein from three independent Western blot analyses of treated cells.

**Figure 3 molecules-25-03268-f003:**
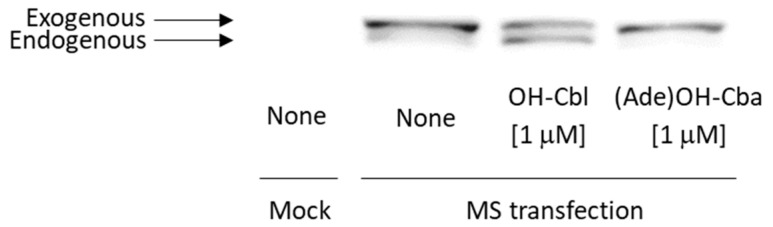
Western blotting of MS protein in MS cDNA-transfected cells grown with or without OH-Cbl and (Ade)OH-Cba. Cells were cultured in the medium with or without 1 µM OH-Cbl or 1 µM (Ade)OH-Cba for 2 days. Other procedures were done as shown in Figure 2. Immunoreactive bands with higher and lower molecular weights are derived from exogenous and endogenous MS proteins, respectively. Data are typical immunoreactive patterns of the MS protein from three independent Western blot analyses of treated cells.

**Figure 4 molecules-25-03268-f004:**
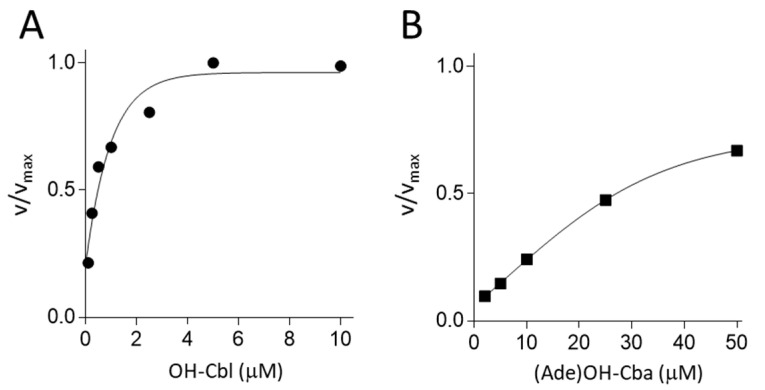
Effects of various concentrations of OH-Cbl (**A**) or (Ade)OH-Cba (**B**) on MS activity in MS cDNA-transfected cells. A cell homogenate of MS cDNA-transfected cells grown without OH-Cbl was used as an apo-MS preparation. MS activity was assayed in the reaction mixture containing various concentrations of OH-Cbl (0.1, 0.25, 0.5, 1, 2.5, 5, and 10 μM) or (Ade)OH-Cba (2, 5, 10, 25, and 50 μM).

**Table 1 molecules-25-03268-t001:**
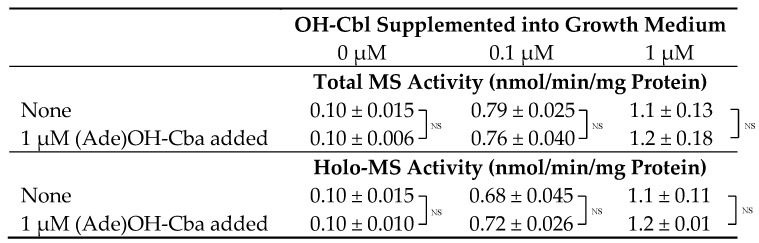
Effects of addition of (Ade)OH-Cba on the total and holo-MS activity in cells grown with or without OH-Cbl.

Cells were cultured in medium containing OH-Cbl at 0, 0.1, or 1 µM (apart from the endogenous Cbl = 10 pM) in the absence or presence of 1 µM (Ade)OH-Cba for 2 days. Total and holo-MS activity in cells was determined with and without the addition of 50 µM OH-Cbl into the reaction mixture, respectively. Data presented as mean ± SD (*n =* 3). No significant difference (NS) was observed between cells cultured in with or without (Ade)OH-Cba when the concentration of Cbl in the medium was the same.

**Table 2 molecules-25-03268-t002:**
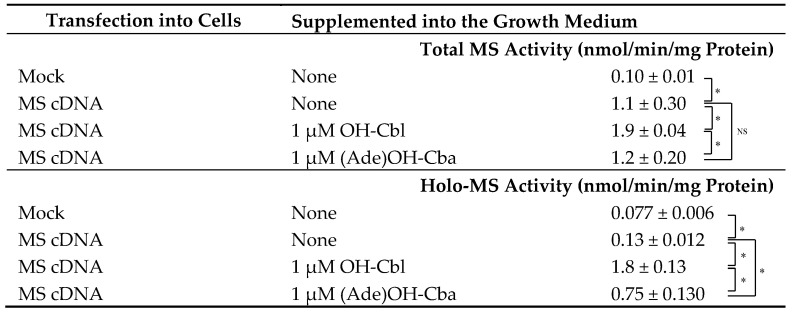
Total and holo-MS activity in MS cDNA-transfected cells grown with or without OH-Cbl and (Ade)OH-Cba.

MS cDNA- or mock-transfected cells were cultured in the absence or presence of 1 µM OH-Cbl or (Ade)OH-Cba for 2 days. Total and holo-MS activities in cells were determined with and without the addition of OH-Cbl into the reaction mixture, respectively. Data presented as mean ± SD (*n =* 3). Asterisk (*) denotes significant differences (*p* < 0.05). No significant difference (NS) was observed between total MS activity in MS cDNA-transfected cells cultured with or without (Ade)OH-Cba.

**Table 3 molecules-25-03268-t003:**
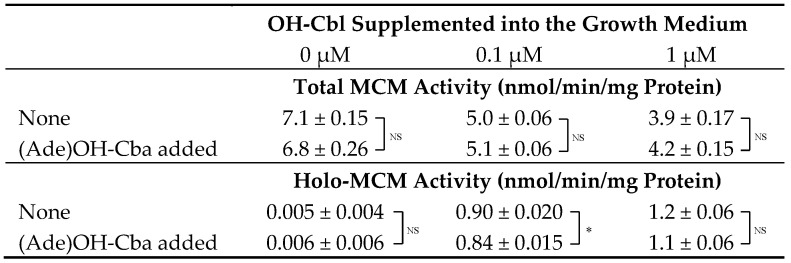
Effects of addition of (Ade)OH-Cba on the total and holo-MCM activity in cells grown in the presence or absence of OH-Cbl.

Cells were cultured in medium containing OH-Cbl at 0, 0.1, or 1 µM (apart from the endogenous Cbl = 10 pM) with or without 1 µM (Ade)OH-Cba for 2 days. The total and holo-methylmalonyl-CoA mutase (MCM) activities in the cells were determined in the presence and absence of 33.3 µM AdoCbl in the reaction mixture, respectively. Data presented as mean ± SD (*n =* 3). Asterisk (*) denotes significant differences (*p* < 0.05). NS represents no significant difference.

**Table 4 molecules-25-03268-t004:**
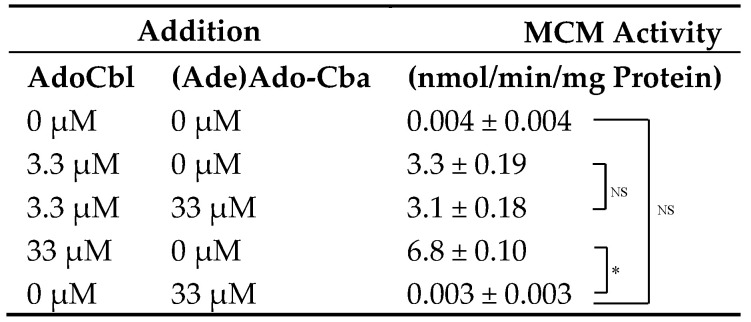
Effects of (Ade)Ado-Cba on the activity of apo-MCM in mammalian cells.

Cells were cultured without supplementation of OH-Cbl for 2 days. MCM activity was determined in the presence of AdoCbl and/or (Ade)Ado-Cba at the indicated concentrations. Data presented as mean ± SD (*n =* 3). Asterisk (*) denotes significant differences (*p* < 0.05). NS denotes no significant difference.

**Table 5 molecules-25-03268-t005:**
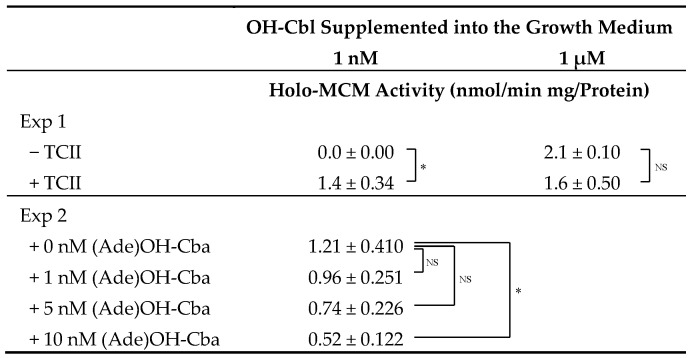
Effects of addition of (Ade)OH-Cba on the activity of holo-MCM in cells grown in the presence of TCII and OH-Cbl.

Exp 1: Cells were cultured together with or without TCII (1 nM) in medium containing 1 nM or 1 µM OH-Cbl for 2 days. Holo-MCM activity in cells was determined without the addition of AdoCbl into the reaction mixture. Results are presented as mean ± SD (*n* = 4). *: *p* < 0.05 versus the values for cells cultured with 1 nM OH-Cbl under the same conditions for TCII. Exp 2: Cells were cultured together with TCII (1 nM) in medium containing OH-Cbl (1 nM) and various concentrations of (Ade)OH-Cba for 2 days. Data presented as mean ± SD (*n* = 3). *: *p* < 0.05 versus control (0 nM (Ade)OH-Cba). NS denotes no significant difference.

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
