# Peer review of "Biological Activity of Pseudovitamin B12 on Cobalamin-Dependent Methylmalonyl-CoA Mutase and Methionine Synthase in Mammalian Cultured COS-7 Cells"

_molecules, 2020, doi:10.3390/molecules25143268_

Round 1

Reviewer 1 Report

It is noticiable the work done by the authors to improve the manuscript.

My general recommendation is to accept the article. However, I advise the authors they modify references 25 and 26 since those references do not describe the methods used in this paper in relation to the estimation of Kd.

Author Response

Thank you very much for your decision letter of 24th, June 2020, with regard to our manuscript (Molecules-842359) with the comments from yourself. We appreciate the comments, which are very helpful. We have tried to revise the manuscript in line with suggestions.

In response to comments from reviewer 1, the following changes were made (as marked in blue).

It is noticiable the work done by the authors to improve the manuscript.

My general recommendation is to accept the article. However, I advise the authors they modify references 25 and 26 since those references do not describe the methods used in this paper in relation to the estimation of Kd.

Ans; The methods used have been described in a revised manuscript (Lines 313-317) according to the reviewer’s suggestion. References 25 and 26 have been changed to reference 31.

Reviewer 2 Report

General comments

The submitted manuscript of Bito et al “Biological Activity of Pseudovitamin B12 on Cobalamin-Dependent Methylmalonyl-CoA Mutase and Methionine Synthase in Mammalian Cultured COS-7 Cells” is devoted to an interesting subject, related to the activity of mammalian B12-dependent enzymes, is using a B12-analogue instead of the “true cofactor”. Such experiment has never been performed on a cell culture and deserves an attention. In addition, the manuscript shows examination of both B12-dependent enzymes, which is an advantage. Unfortunately, the authors either are not familiar with the literature existing on this subject or prefer to ignore it, because a number of previous observations directly contradict some results and inferences of Bito et al. This is not a good habit, and many parts of the submitted manuscript must be rewritten to accommodate discussion / corrections, concerning the subjects outline in my comments (see below). Introduction and abstract are written in a reasonably good English, while the section Results and Discussion is abound in various language mistakes. Numerous corrections are suggested directly within the manuscript file. I suggest a major revision of the submitted work, which has to address the raised issues.

Major comments

(1) Lines 61-62. “Cbl is synthesized only by certain bacteria and is primarily found in predators located higher in the food chain”. The authors leave all readers in total bewilderment concerning appearance of Cbl in predators. Yet, the way of Cbl from microorganisms to animal tissues is well established: (i) bacteria inhabit the digestive system of herbivores and ferment cellulose; (ii) these bacteria produce Cbl; (iii) bacteria die and release Cbl; (iv) herbivores absorb the liberated Cbl and accumulate it in the tissues; (v) predators and omnivores obtain Cbl by consuming herbivores or/and via coprophagy. See, e.g. a review doi: 10.1007/978-94-007-2199-9_18 and references thereof.

(2) Lines 63-64. “Various methanogenic bacteria contain a substantial amount of adenyl cobamide [(Ade)Cba] known as pseudovitamin B12.” The provided references state nothing like that. These microorganisms are methanogenic archaea (not bacteria), and they contain mostly Me-forms of various corrinoids, because the main metabolic pathway in these microorganisms (Wood–Ljungdahl pathway) requires synthesis and transfer of Me-group (involving MeCba). Remove “methanogenic” from the sentence in question.

(3) Lines 69-76. The whole logic of this paragraph should be changed (including Figure 2). It is known that infusion of large quantities of pseudo B12 (2 μg/h over 14 days) into rats is of no consequence for Cbl-related metabolism (doi: 10.1172/JCI115148). Such dose corresponds to ≈ 400 μg/h (10 mg/day !!!) in humans. Therefore, pseudo B12 of natural bacterial origin cannot be harmful for humans, irrespective of the rate of possible bacterial translocation. I would suggest a change of logic to e.g. "limited/contradictory knowledge about Cbl-specificity of mammalian MS"

(4) Lines 78-79. “However, there is little information available regarding the biological activity of (Ade)Cba in mammalian cells.” In fact, there are several very relevant publications devoted exactly to this subject, e.g. Kolhouse et al, Mechanism of Conversion of Human Apo- To Holomethionine Synthase by Various Forms of Cobalamin. J Biol Chem. 1991 Dec 5;266(34):23010-5; Stabler et al, Inhibition of Cobalamin-dependent Enzymes by Cobalamin Analogues in Rats, doi: 10.1172/JCI115148; Yamada et al, Heterologous High Level Expression, Purification, and

Enzymological Properties of Recombinant Rat Cobalamin-dependent Methionine Synthase, doi: 10.1074/jbc.274.50.35571; plus a recent (but not reviewed) bioRxiv publication doi: 10.1101/2020.03.20.997551. A good scientific practice implies citation of at least the peer reviewed publications.

(5) Lines 138-141. “The apparent Kd values for OH-Cbl and (Ade)OH-Cba were estimated to be 0.4 and 26.3 μM, respectively (Figure 5), indicating that (Ade)OH-Cba has significantly lower affinity (approximately 1/65 times) for MS than OH-Cbl.” This result directly contradicts Kolhouse et al (Mechanism of Conversion … PMID: 1744096). These authors detected a half-activation of apo-MS from rat at 1 μM HOCbl and 0.02 μM of [Ade]OH-Cba (pseudo B12), implying that affinity for the latter ligand is much higher. In other words, the apparent dissociation constants, presented by the submitted manuscript and Kolhouse et al, have a 1000-fold difference !!! Bito et al should address this issue and possibly compare the composition of reaction medium, the corrinoid-reducing agents, etc. This is known that maintenance of MS-activity (if using Ti(III) as reducing agent) requires presence of adenosylmethionine (AdoMet), see Yamada et al, Fig. 4, doi: 10.1074/jbc.274.50.35571. As far as I could gather, Bito et al use Ti(III)-reducing system without AdoMet and determine the reaction rate from a single measurement at 30 min. Could it be possible that MS in complex with (Ade)OH-Cba becomes inactivated in several minutes and the submitted manuscript misses most of MS-(Ade)OH-Cba activity?

(6) Lines 233-234. “Considering that the addition of (Ade)Cba had no effect on MCM activity in the absence of TCII, a substantial amount of (Ade)Cba appears to inhibit the TCII-mediated Cbl uptake.” The authors should discuss their result in comparison to Stabler et al. (doi: 10.1172/JCI115148), who demonstrated that infusion of large quantities of (Ade)Cba into blood of rats (containing TCII) neither affected the activity of MCM in liver, nor significantly increased MMA in blood (Table I), indicating absence of Cbl-deficiency.

(7) Lines 236-240. It seems to me that this section should be totally rewritten, taking into account the data from refs., mentioned in the comment (4).

(8) Lines 244-246. “Although it has been reported that (Ade)Cba could bind to TCII at a similar affinity to Cbl [14], only a substantial amount of (Ade)Cba was able to inhibit the TCII-mediated Cbl uptake at a physiological trace concentration of Cbl.” In fact, Cbl and pseudo B12 ([Ade]Cba) have a 50-fold difference in their Kd in favor of Cbl, as follows from Table 4 in ref. (doi: 10.1021/bi062063l). Therefore, there is no wonder that Cbl binds better to TC, than [Ade]Cb.

(9) Lines 254-255. “If large amounts of (Ade)Cba exist in the blood due to bacterial translocation, it might inhibit the TCII-mediated Cbl uptake to accelerate the development of Cbl deficiency.” This statement does not reflect the literature for an in vivo experiment with [Ade]Cba, see doi: 10.1172/JCI115148.

Minor comments

(10) All results. The authors add Cbl to both cell the culture medium and the enzyme reaction medium (depending on the experiment). Therefore, they should always specify to what medium Cbl was added to avoid misunderstanding in statements like “… the addition of 1 μM OH-Cbl significantly increased …” (addition to what?).

(11) Lines 96-97. “Total (sum of holo- and apoenzymes) MS activity was detected at similar levels of the holo-enzyme activity …” The sentence is not clear. Do you mean “Total MS activity was measured with excessive Cbl (what form?) added to cell homogenates (reaction mixtures?). It roughly corresponded to the original holo-enzyme activity (without added Cbl), indicating that the majority of MS existed as a holo-enzyme, while apo-enzyme was generally absent in the cells.” By the way, this was also found in doi: 10.1074/jbc.270.33.19246.

(12) Lines 119-120. “Figure 4 shows the effect of MS protein levels on the addition of OH-Cbl or (Ade)OH-Cba in MS cDNA-transfected cells.” The current sentence means that MS determins Cbl, which makes no sense. I suppose that the correct text should run as follows: “Figure 4 shows the effects of cDNA transfection and addition of OH-Cbl or (Ade)OH-Cba on MS expression in the cells”?

(13) Lines 163-164 and Table 2. “Different superscript letters denote significantly different values (p < 0.05).” Values are different from what (????): mock vs additive, all vs zero, something else? What is the meaning of a, b and c (different significance levels)? The meaning of a, b, c must be specified.

(14) Line 185. I would recommend to add an introducing sentence between the sections about MS and MCM enzymes, e.g. “A similar set of experiments was performed when analyzing activity of another Cbl-dependent enzyme MCM”.

(15) Lines 386-387. “Data represented in Tables 1 and 3 and Exp 1 of Table 5 were statistically analyzed by two-way ANOVA …” It seems to me that analysis by ANOVA has no meaning, because it compares all the datasets with an expected global mean in an attempt to find deviation from the later. Yet, no global mean can be expected for the samples in question. A much more meaningful analysis would cover pairwise comparisons of a blank sample (no Cbl or [Ade]Cba) with the ligand-containing samples, as well as pairwise analysis of Cbl vs [Ade]Cba (added the same concentration to the cells).

Author Response

Thank you very much for your decision letter of 24th, June 2020, with regard to our manuscript (Molecules-842359) with the comments from yourself. We appreciate the comments, which are very helpful. We have tried to revise the manuscript in line with suggestions.

In response to comments from reviewer 2, the following changes were made (as marked in yellow).

Major comments

(1) Lines 61-62. “Cbl is synthesized only by certain bacteria and is primarily found in predators located higher in the food chain”. The authors leave all readers in total bewilderment concerning appearance of Cbl in predators. Yet, the way of Cbl from microorganisms to animal tissues is well established: (i) bacteria inhabit the digestive system of herbivores and ferment cellulose; (ii) these bacteria produce Cbl; (iii) bacteria die and release Cbl; (iv) herbivores absorb the liberated Cbl and accumulate it in the tissues; (v) predators and omnivores obtain Cbl by consuming herbivores or/and via coprophagy. See, e.g. a review doi: 10.1007/978-94-007-2199-9_18 and references thereof.

Ans; Some sentences have been rewritten (Lines 61-63) according to the reviewer’s suggestion.

(2) Lines 63-64. “Various methanogenic bacteria contain a substantial amount of adenyl cobamide [(Ade)Cba] known as pseudovitamin B12.” The provided references state nothing like that. These microorganisms are methanogenic archaea (not bacteria), and they contain mostly Me-forms of various corrinoids, because the main metabolic pathway in these microorganisms (Wood–Ljungdahl pathway) requires synthesis and transfer of Me-group (involving MeCba). Remove “methanogenic” from the sentence in question.

Ans; The word “methanogenic” has been deleted according to the reviewer’s suggestion (Line 65).

(3) Lines 69-76. The whole logic of this paragraph should be changed (including Figure 2). It is known that infusion of large quantities of pseudo B12 (2 μg/h over 14 days) into rats is of no consequence for Cbl-related metabolism (doi: 10.1172/JCI115148). Such dose corresponds to ≈ 400 μg/h (10 mg/day !!!) in humans. Therefore, pseudo B12 of natural bacterial origin cannot be harmful for humans, irrespective of the rate of possible bacterial translocation. I would suggest a change of logic to e.g. "limited/contradictory knowledge about Cbl-specificity of mammalian MS"

Ans; Logic of the paragraph has been changed and rewritten (Lines 70-78) according to the reviewer’s suggestion. Figure 2 has been deleted.

(4) Lines 78-79. “However, there is little information available regarding the biological activity of (Ade)Cba in mammalian cells.” In fact, there are several very relevant publications devoted exactly to this subject, e.g. Kolhouse et al, Mechanism of Conversion of Human Apo- To Holomethionine Synthase by Various Forms of Cobalamin. J Biol Chem. 1991 Dec 5;266(34):23010-5; Stabler et al, Inhibition of Cobalamin-dependent Enzymes by Cobalamin Analogues in Rats, doi: 10.1172/JCI115148; Yamada et al, Heterologous High Level Expression, Purification, and

Enzymological Properties of Recombinant Rat Cobalamin-dependent Methionine Synthase, doi: 10.1074/jbc.274.50.35571; plus a recent (but not reviewed) bioRxiv publication doi: 10.1101/2020.03.20.997551. A good scientific practice implies citation of at least the peer reviewed publications.

Ans; The sentence pointed out by the reviewer has been deleted. New sentences have been added at Lines 70-78 according to the reviewer’s suggestion.

(5) Lines 138-141. “The apparent Kd values for OH-Cbl and (Ade)OH-Cba were estimated to be 0.4 and 26.3 μM, respectively (Figure 5), indicating that (Ade)OH-Cba has significantly lower affinity (approximately 1/65 times) for MS than OH-Cbl.” This result directly contradicts Kolhouse et al (Mechanism of Conversion … PMID: 1744096). These authors detected a half-activation of apo-MS from rat at 1 μM HOCbl and 0.02 μM of [Ade]OH-Cba (pseudo B12), implying that affinity for the latter ligand is much higher. In other words, the apparent dissociation constants, presented by the submitted manuscript and Kolhouse et al, have a 1000-fold difference !!! Bito et al should address this issue and possibly compare the composition of reaction medium, the corrinoid-reducing agents, etc. This is known that maintenance of MS-activity (if using Ti(III) as reducing agent) requires presence of adenosylmethionine (AdoMet), see Yamada et al, Fig. 4, doi: 10.1074/jbc.274.50.35571. As far as I could gather, Bito et al use Ti(III)-reducing system without AdoMet and determine the reaction rate from a single measurement at 30 min. Could it be possible that MS in complex with (Ade)OH-Cba becomes inactivated in several minutes and the submitted manuscript misses most of MS-(Ade)OH-Cba activity?

Ans; We confirmed that there is no problem on MS assay conditions. Some sentences have been added at Lines 248-257 to discuss such contradiction.

(6) Lines 233-234. “Considering that the addition of (Ade)Cba had no effect on MCM activity in the absence of TCII, a substantial amount of (Ade)Cba appears to inhibit the TCII-mediated Cbl uptake.” The authors should discuss their result in comparison to Stabler et al. (doi: 10.1172/JCI115148), who demonstrated that infusion of large quantities of (Ade)Cba into blood of rats (containing TCII) neither affected the activity of MCM in liver, nor significantly increased MMA in blood (Table I), indicating absence of Cbl-deficiency.

Ans; Some sentences have been added at Lines 265-278 to discuss the results.

(7) Lines 236-240. It seems to me that this section should be totally rewritten, taking into account the data from refs., mentioned in the comment (4).

Ans; This section has been rewritten according to the reviewer’s suggestion (Lines 242-248).

(8) Lines 244-246. “Although it has been reported that (Ade)Cba could bind to TCII at a similar affinity to Cbl [14], only a substantial amount of (Ade)Cba was able to inhibit the TCII-mediated Cbl uptake at a physiological trace concentration of Cbl.” In fact, Cbl and pseudo B12 ([Ade]Cba) have a 50-fold difference in their Kd in favor of Cbl, as follows from Table 4 in ref. (doi: 10.1021/bi062063l). Therefore, there is no wonder that Cbl binds better to TC, than [Ade]Cb.

Ans; This section has been rewritten according to the reviewer’s suggestion (Lines 242-261).

(9) Lines 254-255. “If large amounts of (Ade)Cba exist in the blood due to bacterial translocation, it might inhibit the TCII-mediated Cbl uptake to accelerate the development of Cbl deficiency.” This statement does not reflect the literature for an in vivo experiment with [Ade]Cba, see doi: 10.1172/JCI115148.

Ans; This section has been rewritten according to the reviewer’s suggestion (Lines 242-261).

Minor comments

(10) All results. The authors add Cbl to both cell the culture medium and the enzyme reaction medium (depending on the experiment). Therefore, they should always specify to what medium Cbl was added to avoid misunderstanding in statements like “… the addition of 1 μM OH-Cbl significantly increased …” (addition to what?).

Ans; We have tried to revise a manuscript in the line of the reviewer’s suggestion.

(11) Lines 96-97. “Total (sum of holo- and apoenzymes) MS activity was detected at similar levels of the holo-enzyme activity …” The sentence is not clear. Do you mean “Total MS activity was measured with excessive Cbl (what form?) added to cell homogenates (reaction mixtures?). It roughly corresponded to the original holo-enzyme activity (without added Cbl), indicating that the majority of MS existed as a holo-enzyme, while apo-enzyme was generally absent in the cells.” By the way, this was also found in doi: 10.1074/jbc.270.33.19246.

Ans; The sentences have been revised according to the reviewer’s suggestion (Lines 91-94).

(12) Lines 119-120. “Figure 4 shows the effect of MS protein levels on the addition of OH-Cbl or (Ade)OH-Cba in MS cDNA-transfected cells.” The current sentence means that MS determins Cbl, which makes no sense. I suppose that the correct text should run as follows: “Figure 4 shows the effects of cDNA transfection and addition of OH-Cbl or (Ade)OH-Cba on MS expression in the cells”?

Ans; The sentence has been revised according to the reviewer’s suggestion (Lines 124-125).

(13) Lines 163-164 and Table 2. “Different superscript letters denote significantly different values (p < 0.05).” Values are different from what (????): mock vs additive, all vs zero, something else? What is the meaning of a, b and c (different significance levels)? The meaning of a, b, c must be specified.

Ans; Appearances of all Tables have revised according to the reviewer’s suggestion.  

(14) Line 185. I would recommend to add an introducing sentence between the sections about MS and MCM enzymes, e.g. “A similar set of experiments was performed when analyzing activity of another Cbl-dependent enzyme MCM”.

Ans; A sentence has been added at Lines 179-180 according to the reviewer’s suggestion.

(15) Lines 386-387. “Data represented in Tables 1 and 3 and Exp 1 of Table 5 were statistically analyzed by two-way ANOVA …” It seems to me that analysis by ANOVA has no meaning, because it compares all the datasets with an expected global mean in an attempt to find deviation from the later. Yet, no global mean can be expected for the samples in question. A much more meaningful analysis would cover pairwise comparisons of a blank sample (no Cbl or [Ade]Cba) with the ligand-containing samples, as well as pairwise analysis of Cbl vs [Ade]Cba (added the same concentration to the cells).

Ans; The method of statistical analysis has been changed according to the reviewer’s suggestion. Some sentences have been added at Lines 403-405 according to the reviewer’s suggestion.

Additional comments described on the manuscript PDF

Ans; All comments have been revised in red according to the reviewer’s suggestions.

Round 2

Reviewer 2 Report

General comments

The revised version of manuscript “Biological Activity of Pseudovitamin B12 on Cobalamin-Dependent Methylmalonyl-CoA Mutase and Methionine Synthase in Mammalian Cultured COS-7 Cells” by Bito et al presents a considerably improved report about the performed work. A number of minor suggestions concern corrections of the language, clarifying remarks and one possibly relevant explanation of the observed contradiction between the current work and ref. 18. Most of the corrections are indicated within the submitted file. The manuscript of Bito et al can be published after minor revision. No further proofreading is required from my side.

Minor comments

(1) Line 257. If the authors find this suitable, I suggest to add the following text concerning the observed contradiction with ref. 18. “During enzymatic reactions, we reduced Cbl / Cba by titanium citrate, which usually gives [Co1+] as the predominant cofactor form in the reaction medium. In contrast, Kolhouse at al employed reduction by mercaptoethanol, where the major stady-state form under reduction-oxidation is [Co2+]. The oxidative state of Co-ion inside the corrin ring affects coordination patterns of all corrinoids in respect to the surrounding ligands, which might account for the reported differences. Further studies are needed to clarify this issue.”

(2) See remarks in the file.

Author Response

15th July 2020

Dear Reviewer 2,

Thank you very much for your decision letter of 14th, July 2020, with regard to our manuscript (Molecules-842359) with the comments from yourself. We appreciate the comments, which are very helpful. We have tried to revise the manuscript in line with suggestions.

In response to comments from reviewer 2, the following changes were made (as marked in yellow).

The revised version of manuscript “Biological Activity of Pseudovitamin B12 on Cobalamin-Dependent Methylmalonyl-CoA Mutase and Methionine Synthase in Mammalian Cultured COS-7 Cells” by Bito et al presents a considerably improved report about the performed work. A number of minor suggestions concern corrections of the language, clarifying remarks and one possibly relevant explanation of the observed contradiction between the current work and ref. 18. Most of the corrections are indicated within the submitted file. The manuscript of Bito et al can be published after minor revision. No further proofreading is required from my side.

Minor comments

(1) Line 257. If the authors find this suitable, I suggest to add the following text concerning the observed contradiction with ref. 18. “During enzymatic reactions, we reduced Cbl / Cba by titanium citrate, which usually gives [Co1+] as the predominant cofactor form in the reaction medium. In contrast, Kolhouse at al employed reduction by mercaptoethanol, where the major stady-state form under reduction-oxidation is [Co2+]. The oxidative state of Co-ion inside the corrin ring affects coordination patterns of all corrinoids in respect to the surrounding ligands, which might account for the reported differences. Further studies are needed to clarify this issue.”

Ans: Sentences have been added at Lines 266-272 according to the reviewer’s suggestion.

(2) See remarks in the file.

Ans: All comments have been revised according to the reviewer’s suggestions (the manuscript PDF file).

I hope that you will be able to consider this revised manuscript for possible publication in molecules.

Best Regards,

Tomohiro Bito

This manuscript is a resubmission of an earlier submission. The following is a list of the peer review reports and author responses from that submission.

Round 1

Reviewer 1 Report

Bito, T. et al

Biological Activity of Pseudovitamin B12 on Cobalamin-Dependent Methylmalonyl-CoA Mutase and Methionine Synthase in Mammalian Cultured COS-7 Cells

The present manuscript addresses an interesting question derived from past observations on the availability of bacteria derived vitamins: whether pseudo-vitamin B12 can interfere or substitute  for cobalamin into mammalian methionine synthase and methylmalonyl-CoA mutase. Although the work is for the most part correctly done, there are a number of issues that, in the opinion of this referee, should be added or modified. All of them, refer to the first enzyme studied (methionine synthase):

1.- From the data they present on figure 3, it is clear that the polypeptide levels of MS are greatly influenced by the presence of cbl in the growth medium, and that (Ade)Cba does not induce accumulation of this polypeptide. Rather, it remains as in the case of no additions. The authors explain that OH-Cbl induces translation of the MS mRNA. In table 2, they observe that the holoenzyme activity of (Ade)Cba-treated cells is greater than untreated and lower than OH-Cbl-treated cells, when looking at the total enzyme, they observe a similar value for (Ade)Cba-treated and untreated cells and this is lower than that from OH-Cbl-treated cells. The authors rightly suggest that many of these differences may be due to differences in polypeptide levels while others are due to lower catalytic power. However, in the absence of mesurements of protein levels, all those suggestions are speculative. It would be a very wellcome and clarifying addition to show some western blots. That way, it could be discerned what effects are due to differences in actual enzyme catalytic performance and which ones to differences in polypeptide levels.

2.- The study makes extensive use of enzyme assays done at saturating concentrations of substrate. This is a good and sensible initial approach. However, if the authors want to sustain the conclusion that (Ade)Cba "has a significantly lower affinity for MS than Cbl" (Line 214), it is mandatory to show kinetic analysis and/or studies of holoenzyme formation. The authors report Km values for OH-Cbl and (Ade)Cba but there is no plot sustaining those values, they have no standard error and no mention in the text on how those values were estimated. Also, being cobalamine a co-factor, it seems rather difficult to provide a Km value for it as if it was a substrate (the enzyme, once cobalamine is bound, does not easily release it, it becomes part of the holoenzyme; Km, on its turn, depends greatly on the existence of a reversible binding step).

Apart from those two issues, I found the work very well writen and probably very interesting for the readership

Author Response

20th May 2020

Dear Reviewer 1:

 Thank you very much for your decision letter of 10th, May, 2020, with regard to our manuscript (Molecules-801971) with the comments from yourself. We appreciate the comments, which are very helpful. We have tried to revise the manuscript in line with suggestions.

In response to comments from yourself, the following changes were made (as marked in green).

1) From the data they present on figure 3, it is clear that the polypeptide levels of MS are greatly influenced by the presence of cbl in the growth medium, and that (Ade)Cba does not induce accumulation of this polypeptide. Rather, it remains as in the case of no additions. The authors explain that OH-Cbl induces translation of the MS mRNA. In table 2, they observe that the holoenzyme activity of (Ade)Cba-treated cells is greater than untreated and lower than OH-Cbl-treated cells, when looking at the total enzyme, they observe a similar value for (Ade)Cba-treated and untreated cells and this is lower than that from OH-Cbl-treated cells. The authors rightly suggest that many of these differences may be due to differences in polypeptide levels while others are due to lower catalytic power. However, in the absence of measurements of protein levels, all those suggestions are speculative. It would be a very welcome and clarifying addition to show some western blots. That way, it could be discerned what effects are due to differences in actual enzyme catalytic performance and which ones to differences in polypeptide levels.

Ans:

Figure 4, the legend of Figure 4, and the sentences associated with Figure 4 have been added at page 5, lines 167-175 and page 4, lines 118-126 in a revised manuscript (green marker) according to the suggestions.

2) The study makes extensive use of enzyme assays done at saturating concentrations of substrate. This is a good and sensible initial approach. However, if the authors want to sustain the conclusion that (Ade)Cba "has a significantly lower affinity for MS than Cbl" (Line 214), it is mandatory to show kinetic analysis and/or studies of holoenzyme formation. The authors report Km values for OH-Cbl and (Ade)Cba but there is no plot sustaining those values, they have no standard error and no mention in the text on how those values were estimated. Also, being cobalamine a co-factor, it seems rather difficult to provide a Km value for it as if it was a substrate (the enzyme, once cobalamine is bound, does not easily release it, it becomes part of the holoenzyme; Km, on its turn, depends greatly on the existence of a reversible binding step).

Apart from those two issues, I found the work very well writen and probably very interesting for the readership

Ans:

Figure 5 and the legend of Figure 5 have been added at page 6, lines 176-179 in a revised manuscript (green marker) according to the suggestions.

Reviewer 2 Report

There are several reports on the prevalence of Vitamin B12 deficiency specially in the elderly, which is one more reason that makes this study interesting in our aging society. Nevertheless, there are controversies in the literature regarding the use of in vitro studies as the most appropriate approach to examine physiological concentrations of micronutrients as in the case of Vitamin B12.

Overall this is an interesting report that addresses an important issue on the possible impact and efficacy of diet supplements containing pseudo vitamin B12 as a suitable and alternative source of Vitamin B12 or even for assessing the capability of these molecules as functional co-factors of B12 depending enzymes ultimately leading to a better understanding of the impact of some of these natural occurring  B12 analogues might have in the human B12 metabolism.

This manuscript is well written and clear although some clarifications and corrections need to be addressed.

Major comments:

In figure 1 the list of structures and legend still lack a better and clear correspondence to the abbreviated designation used in the text. This should be included in the figure or legend.

In terms of presentation, the tables described in the results section, should be displayed in a more consistent form. For example, table 1 shows the Hole-MS activity as a first line and the order is reversed in table 2 and 3. This should be homogenised.

In terms of experimental design and methods the authors should clarify if in the cell culture experiments there is no exogenous contribution  source of Vitamin B12 that could be present either in the cell culture media or in the FBS used as supplementation. This is an in vitro study and there is evidence that certain micronutrients are present in several commercially available cell culture media at concentrations sometimes higher than those found in human serum.

In terms of methods could the authors explain the reason for the selection of one unique concentration of AdeOH-Cba for most experiments performed, namely in table 1, 2 and 3? Was there a prior dose dependent study or information?

Minor comments:

In line 86-88 the text seems to be repeated and incorrect.

In line 124 Table 1 seems to be incorrectly cited.

Author Response

20th May 2020

Dear Reviewer 2:

 Thank you very much for your decision letter of 10th, May, 2020, with regard to our manuscript (Molecules-801971) with the comments from yourself. We appreciate the comments, which are very helpful. We have tried to revise the manuscript in line with suggestions.

In response to comments from yourself, the following changes were made (as marked in yellow).

1) In figure 1 the list of structures and legend still lack a better and clear correspondence to the abbreviated designation used in the text. This should be included in the figure or legend.

Ans: Figure 1 and the legend of Figure 1 have been revised at page 2, lines 48-51 in a revised manuscript (yellow marker) according to the suggestions.

2) In terms of presentation, the tables described in the results section, should be displayed in a more consistent form. For example, table 1 shows the Hole-MS activity as a first line and the order is reversed in table 2 and 3. This should be homogenised.

Ans: Table 1 has been revised at page 4 in a revised manuscript (yellow marker) according to the suggestions.

3) In terms of experimental design and methods the authors should clarify if in the cell culture experiments there is no exogenous contribution source of Vitamin B12 that could be present either in the cell culture media or in the FBS used as supplementation. This is an in vitro study and there is evidence that certain micronutrients are present in several commercially available cell culture media at concentrations sometimes higher than those found in human serum.

Ans: Sentences have been added at page 3, lines 92-93 in a revised manuscript (yellow marker) according to the suggestions.

4) In terms of methods could the authors explain the reason for the selection of one unique concentration of AdeOH-Cba for most experiments performed, namely in table 1, 2 and 3? Was there a prior dose dependent study or information?

Ans: Sentence has been added at page 3, lines 98-99 in a revised manuscript (yellow marker) according to your suggestions.

5) In line 86-88 the text seems to be repeated and incorrect.

Ans: The sentence (at page 3, lines 89-91 in a revised manuscript) is correct. This sentence is consisted from the culture conditions of cell (in the presence or absence of OH-Cbl) and the preparation method of reaction mixture to assay total- or holo-MS activity in cultured cell.

6) In line 124 Table 1 seems to be incorrectly cited.

Ans: The cited has been revised according to your instruction (at page 4, lines 110-112 in a revised manuscript (yellow marker)).

Reviewer 3 Report

The manuscript by Bito T, et al entitled „Biological Activity of Pseudovitamin B12 on Cobalamin-Dependent Methylmalonyl-CoA Mutase and Methionine Synthase in Mammalian Cultured COS-7 Cells “ explores whether a natural vitamin B12 analogue could interfere or substitute B12 activity in cells of  mammalian origin.  Considering the importance of vitamin B12 in general practice, popularity of specific diets and B12 supplementation, the results of current study are of general interest. I have two suggestions for improving the manuscript though.

The text is not easy to follow for someone not working in the field. By taking my time and thinking every sentence through it all made sense, but it is easy to lose track. Perhaps Figure 2 can be improved by a) indication of different origins of Cbl and (Ade)Cba; b) inclusion of holo-and apo MS and MCM and the Cbl/Cba potential role in this change. You may also consider explaining all the abbreviations used in figure 2 in its legend. Alternatively, the study design could be explained in more details in Introduction. I could not find a graphical abstract, but in my imagination an improved version of Figure 2 fits nicely.

For enzyme activity determination methods only a reference to previous work is given. That reference describes measurements of cobalamines, but for MS and MCM enzyme activity measurements are, in my opinion, described in insufficient details and refer to yet another studies. I understand that homocysteine and methylmalonic acid levels were measured by HPLC. In contrary to cobalamines these compounds do not absorb UV-VIS light very well. Was there any derivatization done? More importantly, were the substrate (that are converted into homocysteine or methylmalonic acid) levels controlled in any way or did you relay only on substrates occurring naturally in cell lysates? The manuscript intends to characterize biological activity of these two enzymes and I believe that the key aspects (not necessarily all details) of the assays used to measure these activities should be more easily available than non-open-access references in a reference.

Author Response

20th May 2020

Dear Reviewer 3:

 Thank you very much for your decision letter of 10th, May, 2020, with regard to our manuscript (Molecules-801971) with the comments from yourself. We appreciate the comments, which are very helpful. We have tried to revise the manuscript in line with suggestions.

In response to comments from yourself, the following changes were made (as marked in blue).

1) The text is not easy to follow for someone not working in the field. By taking my time and thinking every sentence through it all made sense, but it is easy to lose track. Perhaps Figure 2 can be improved by a) indication of different origins of Cbl and (Ade)Cba; b) inclusion of holo-and apo MS and MCM and the Cbl/Cba potential role in this change. You may also consider explaining all the abbreviations used in figure 2 in its legend. Alternatively, the study design could be explained in more details in Introduction. I could not find a graphical abstract, but in my imagination an improved version of Figure 2 fits nicely.

Ans:

Sentences have been added at page 2, lines 60-62 and their references (7 and 8) have been added in References section of a revised manuscript (blue marker) according to the suggestions.

In addition, Figure 2 and the legend of Figure 2 have been revised at page 3, lines 83-85 in a revised manuscript (blue marker) according to the suggestions.

2) For enzyme activity determination methods only a reference to previous work is given. That reference describes measurements of cobalamines, but for MS and MCM enzyme activity measurements are, in my opinion, described in insufficient details and refer to yet another studies. I understand that homocysteine and methylmalonic acid levels were measured by HPLC. In contrary to cobalamines these compounds do not absorb UV-VIS light very well. Was there any derivatization done? More importantly, were the substrate (that are converted into homocysteine or methylmalonic acid) levels controlled in any way or did you relay only on substrates occurring naturally in cell lysates? The manuscript intends to characterize biological activity of these two enzymes and I believe that the key aspects (not necessarily all details) of the assays used to measure these activities should be more easily available than non-open-access references in a reference.

Ans:

Sentences have been added at page 9, lines 284-294 and their references (23 and 24) have been added in References section of a revised manuscript (blue marker) according to the suggestions.

Round 2

Reviewer 1 Report

The authors have shown a great diligence in making amendments to the manuscript and, as a result, this has improved noticiably. However, from my point of view, there is an important issue still unresolved. The enzyme kinetics experiments on the association of OH-Cbl and Ade-Cba are an important part in this manuscript since, although they are brief, they provide a sound base from which the authors can build their conclusions on methionine synthase. Unfortunately, in my opinion, those experiments are not correctly approached. This is an issue that was already mentioned in my previous report. On the other hand, they may not be difficult to reconduct.

Being cobalamine a cofactor bound tightly (so much so that under normal circumstances there is no need to add cobalamine in the reaction mixture to assay methionine synthase), to estimate apparent Km and Vmax as if those cofactors were substrates is not correct; Cbls are not binding, being modifed and released as products. Rather, the experiments done could be described as “reconstitution assays” since, as the authors say in the manuscript, only that enzyme expressed in transfected cells responded to exogenous addition of cobalamine forms (due to the overexpression, many enzyme molecules remained as apo-enzymes, as shown on table 2; those expressed normally had incorporated the coenzyme prior to the assay and, as expected, they were not susceptible to exchange it for exogenous molecules). Therefore, Km is not an acceptable cobalamine binding constant and Vmax would only be useful to have a semi-quantitative idea of the amount of newly formed holoenzyme. It could be argued that the experiments shown on Figure 5 showed a linear relationship between 1/[OH-Cbl] or 1/[Ade-OH-Cba] forms and 1/v and that should be an indication that the approach used was correct. However, if we look at a general essential activator velocity equation:

v=Vmax' [S]/(Km(1+(Kx/[X]))+[S](1+(αKx/[X]))); where S is the substrate, X is the activator, Vmax' = βkcat [E0], Km is the Michaelis -Menten constant, and Kx and αKx are the specific and the allosteric dissociation constants for the essential activator, respectively.

We should see that, that upon rearranging and taking reciprocals, the data obeying that expression, when plotted as 1/velocity vs 1/[X] should draw straight lines, but the slope and intercept of those lines will not produce estimates of Km and Vmax.

If the authors want to provide a measure of binding, activation or reconstitution, they should analyse cobalamin forms as tight-binding essential activators or, alternatively, simply as a tight-binding ligands, but not as a substrate. If a simple binding approach is chosen, maybe the dissociation constant Kd can be properly calculated from the same experimental data. It would be enough to use a reasonable threshold to substract the activity of the holoenzymes present at [Cbl]=0 and to fit the data to B/Bmax=[Cbl]/(Kd+[Cbl]), where activity would be a proxy for B. In any case, if the value of Kd happens to lie in the same order of magnitude as the concentration of enzyme used, a Morrison approach would be more appropriate.

That is my sole reservation for an, otherwise, excellent manuscript

Author Response

28th May 2020

Dear Reviewer 1:

 Thank you very much for your decision letter of 23th, May, 2020, with regard to our manuscript (Molecules-801971) with the comments from yourself. We appreciate the comments, which are very helpful. We have tried to revise the manuscript in line with suggestions.

In response to comments from yourself, the following changes were made (as marked in yellow).

The authors have shown a great diligence in making amendments to the manuscript and, as a result, this has improved noticiably. However, from my point of view, there is an important issue still unresolved. The enzyme kinetics experiments on the association of OH-Cbl and Ade-Cba are an important part in this manuscript since, although they are brief, they provide a sound base from which the authors can build their conclusions on methionine synthase. Unfortunately, in my opinion, those experiments are not correctly approached. This is an issue that was already mentioned in my previous report. On the other hand, they may not be difficult to reconduct.

Being cobalamine a cofactor bound tightly (so much so that under normal circumstances there is no need to add cobalamine in the reaction mixture to assay methionine synthase), to estimate apparent Km and Vmax as if those cofactors were substrates is not correct; Cbls are not binding, being modifed and released as products. Rather, the experiments done could be described as “reconstitution assays” since, as the authors say in the manuscript, only that enzyme expressed in transfected cells responded to exogenous addition of cobalamine forms (due to the overexpression, many enzyme molecules remained as apo-enzymes, as shown on table 2; those expressed normally had incorporated the coenzyme prior to the assay and, as expected, they were not susceptible to exchange it for exogenous molecules). Therefore, Km is not an acceptable cobalamine binding constant and Vmax would only be useful to have a semi-quantitative idea of the amount of newly formed holoenzyme. It could be argued that the experiments shown on Figure 5 showed a linear relationship between 1/[OH-Cbl] or 1/[Ade-OH-Cba] forms and 1/v and that should be an indication that the approach used was correct. However, if we look at a general essential activator velocity equation:

v=Vmax' [S]/(Km(1+(Kx/[X]))+[S](1+(αKx/[X]))); where S is the substrate, X is the activator, Vmax' = βkcat [E0], Km is the Michaelis -Menten constant, and Kx and αKx are the specific and the allosteric dissociation constants for the essential activator, respectively.

We should see that, that upon rearranging and taking reciprocals, the data obeying that expression, when plotted as 1/velocity vs 1/[X] should draw straight lines, but the slope and intercept of those lines will not produce estimates of Km and Vmax.

If the authors want to provide a measure of binding, activation or reconstitution, they should analyse cobalamin forms as tight-binding essential activators or, alternatively, simply as a tight-binding ligands, but not as a substrate. If a simple binding approach is chosen, maybe the dissociation constant Kd can be properly calculated from the same experimental data. It would be enough to use a reasonable threshold to substract the activity of the holoenzymes present at [Cbl]=0 and to fit the data to B/Bmax=[Cbl]/(Kd+[Cbl]), where activity would be a proxy for B. In any case, if the value of Kd happens to lie in the same order of magnitude as the concentration of enzyme used, a Morrison approach would be more appropriate.

Ans:

Thank you for your indication and proposal. We appreciate many your comments, which are very helpful for us. I am terribly sorry. We have mistaken how to use the Km value. According to your suggestions, we have tried the calculation of KD value using our data. However, it was too difficult for us to calculate the KD values in this study, because we did not check the molar concentration of methionine synthase using our assay system. I am truly sorry for not being able to use your great idea.

Therefore, we removed the figure of the Lineweaver–Burk plot, the figure legend in a revised manuscript. Sentences have been revised at page 1, lines 26-28 and page 4, lines 130-132 in a revised manuscript (yellow marker) according to your suggestions.

Reviewer 2 Report

The authors have answered positively all my comments. 

Author Response

Reviewer 2;

Thank you very much for your decision letter of 25th, May, 2020, with regard to our manuscript (Molecules-801971) with the comments from yourself.